# Characterization of Humic Substances from Taiga and Tundra Soils by EPR Spectroscopy

**Evgeny Lodygin [1],\***, **Roman Vasilevich [1]** and **Evgeny Abakumov [2]**

[1] Institute of Biology, Komi Science Center, Ural Branch, Russian Academy of Sciences, 167982 Syktyvkar, Russia
[2] Department of Applied Ecology, Faculty of Biology, Saint Petersburg State University, 199178 St. Petersburg, Russia
\* Correspondence: lodigin@ib.komisc.ru

**Abstract:** Paramagnetic activity is a fundamental property of humic substances (HSs). The agricultural use of soils does not only affect the qualitative and quantitative compositions of HSs, but also the content of free radicals (FRs) in their structure. Changes in the composition of incoming plant residues and hydrothermal conditions have an effect on soil humification rates and the paramagnetic properties of humic (HAs) and fulvic acids (FAs). Data on the influence of various factors on the content of FRs in HAs and FAs are not enough. Therefore, the purpose of this study was to examine the influence of the degree of hydromorphism and agricultural use on the paramagnetic properties of HA and FA samples obtained from taiga and tundra soils. Studies have shown that the increased hydromorphism in taiga soils leads to the growing concentration of FRs in the HA molecular structure. HAs in virgin tundra soils exhibit a lower content of unpaired electrons when shifting from automorphic soils to hydromorphic ones. Going from the south to the north, the paramagnetic activity of both HAs and FAs tends to decrease due to the overall reduction of the number of polyconjugated systems in the tundra soil HSs. The comparative analysis of the paramagnetic properties in HAs and FAs of virgin and arable soils revealed that their agricultural use reduces the FR concentration in the structure of HSs, in other words it leads to the accumulation of biothermodynamically stable and more humified compounds in the arable horizons. This contributes to the stabilization of SOM in arable soils.

**Keywords:** humic acids; fulvic acids; paramagnetic activity; free radicals; soil hydromorphism

## 1. Introduction

In recent years, several articles have questioned the role and even existence of soil humic substances (HSs) as a distinct entity in soil organic matter (SOM) [1]. They are dealt with questions of SOM as a continuum of degradation reactions by unspecified processes from the original biomass inputs to carboxylic acids, and eventually to $CO_2$, without the formation of new classes of compounds along the way. However, most reputable researchers in the field of HS chemistry fundamentally disagree with these views and highlight the errors and failure to consider works and evidence that run counter to their pre-conceived views [2–4].

We believe that this complex matter (in both literal and figurative senses) could benefit from a better cooperation between all scientific disciplines devoted to soil studies and propose to consider HSs as a prominent agent of the soil ecosystem, which adequately mirrors the evolution of SOM, depending on the changing environment.

SOM is the most important component of the soil and which mainly determines its fertility. Many articles have been focused on the impact of agricultural soil use on composition and properties of SOM [5–7]. The findings of these studies highlight the degree of transformation of SOM under the influence of plowing, organic and mineral fertilizers and crop cultivation. However, most of these studies are only cut down to the determination of the total amount of carbon and nitrogen or the analysis of the group and

fractional composition of humus. There are significantly fewer articles devoted to the study of transformation of the molecular structure of SOM affected by agriculture.

Numerous experiments studying soil organic matter have proved the immense role of free radicals (FRs) in their semiquinoid form in biochemical processes [8–10]. Basically, FRs are represented by an atom or molecular group with a free unpaired electron which shares its paramagnetic properties with the system, making it possible to trace them with magnetic radiospectroscopic methods. Therefore, in physics, such unpaired electrons are often called paramagnetic centers, and in chemistry—FRs. In inorganic systems, the lifespan of FRs is short if any, but in more complex organic compounds, FRs acquire an unnatural stability, which is, in most cases communicated to them by an electronegative aromatic nucleus [11].

The organic radicals, with their stability associated with the delocalization of an unpaired electron, along the system of conjugated bonds of the polynuclear structure of HS molecules, serve to determine the ability to be naturally paramagnetic for HSs. This property indicates a potential for another approach to study HSs as biopolymers with a natural "spin label". The parameters of "spin labels" (stable free radicals covalently bound to macromolecules) measured by means of the electron paramagnetic resonance, provide information on the structure of conformational dynamics and the microrelief of biological structural elements [12]. Semiquinoid FRs serve as a traditional example of such radicals. According to today's concepts, these radicals are immediate intermediate participants of the stepwise oxidation and polymerization process of humus formation in soils [13–15]. The analyzed results of researchers' work showed that the FR concentration in the HS structure is strongly affected both by various zonal bioclimatic factors [16], and human activity [17,18].

The working hypothesis of our studies was that one of the parameters affecting the FR content in the HS structure is the agricultural use of soils. The results of studies of plowing affecting the molecular composition of HSs in the forest zone soils are extremely contradictory. Thus, according to some researchers [19], in soddy-podzolic and bog-podzolic soils, this leads to an increased proportion of fulvic acids (FAs), in comparison to humic acids (HAs) throughout the profile. The findings of other researchers did not show any significant difference in the composition of HSs in virgin and arable soils [20], while the third group reported an increase in the HA content [21,22]. The research of our own revealed that the agricultural use affects the qualitative and quantitative composition of HSs, while the proportion of strongly acidic carboxyl groups in the composition of FAs decreases [23–25].

Therefore, the purpose of our research was to examine the influence of the degree of hydromorphism and agricultural use on the paramagnetic properties of HA and FA samples obtained from taiga and tundra soils.

## 2. Field Sampling

The research embraced six key soils in taiga (virgin Eutric Albic Histic Retisol, and two Eutric Albic Retisols—virgin and arable ones) and tundra zones (virgin Histic Cryosol, and two Gleyic Stagnosols—virgin and arable ones) (Table 1). The soil types were identified, according to WRB [26]. Figure 1 displays the geography for the samples collected.

**Table 1.** Study objects.

| Soil Name | Vegetation Type | Coordinates |
|---|---|---|
| | Taiga zone | |
| virgin Eutric Albic Histic Retisol | Haircap—sphagnum birch—spruce woodland | 61°40′ N, 50°41′ E |
| virgin Eutric Albic Retisol | Bilberry—green-moss birch—spruce woodland | 61°40′ N, 50°41′ E |
| arable Eutric Albic Retisol | Pea and oat mixture | 61°39′ N, 50°44′ E |
| | Tundra zone | |
| virgin Histic Cryosol | Moss and lichen tundra | 67°35′ N, 64°09′ E |
| virgin Gleyic Stagnosol | Willow—dwarf birch moss tundra | 67°31′ N, 64°07′ E |
| arable Gleyic Stagnosol | Bluegrass and foxtail meadow | 67°31′ N, 64°07′ E |

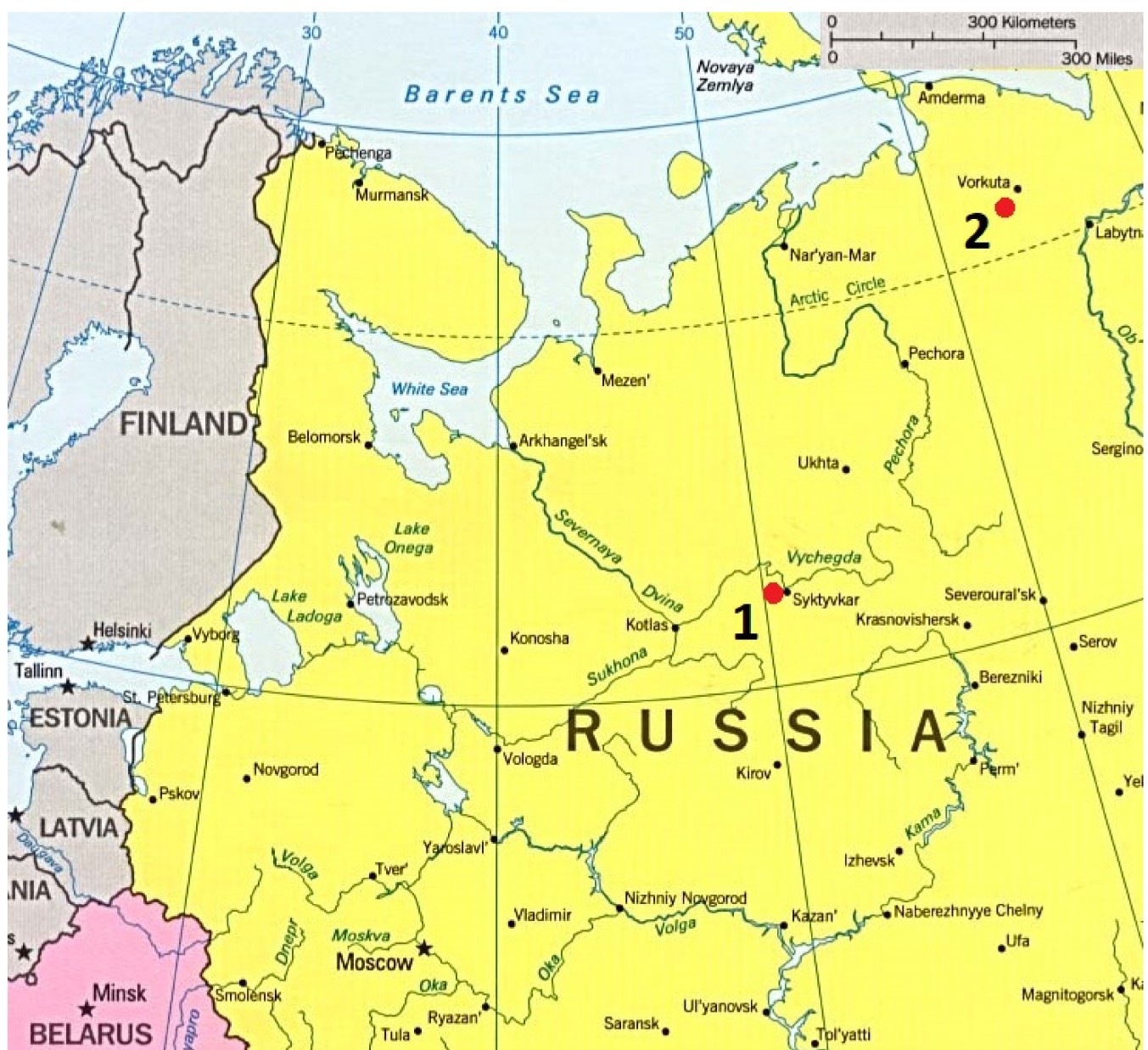

**Figure 1.** Locations of the sampling points of the taiga (1), and the tundra (2) zones.

All Retisols under study are in the taiga zone. Virgin plots of Eutric Albic Retisol and Eutric Albic Histic Retisol were located on the catena. The high position of the catena in the relief made it possible for us to evaluate the impact of the moisture degree on the paramagnetic properties of HSs. The virgin soil plots are located at the Maksimovsky Soil-Ecological Station of the Institute of Biology. The Eutric Albic Retisol plot is eight km to the west of Syktyvkar, on the top of the watershed hill. The plot of Eutric Albic Histic Retisol is 100 m apart from the latter. It is a micro-depression between low and flat hills.

The arable Eutric Albic Retisols plot is on the Syktyvkar state farm fields, 5.6 km to the south-west of Syktyvkar. The relief is flat, with low hills. The development period approximates 40 years. The site operates a closed drainage system.

The tundra soils exposed to the research are located in the Bolshezemelskaya tundra featuring a massive insular distribution of permafrost. Stagnation (gleying) and peat accumulation are typical soil formation processes in the tundra zone [27].

The virgin Gleyic Stagnosol plot is situated on a gradual slope of the Nerusovei-musyur moraine hill. The spot is in a willow and dwarf birch mossy small-hummock tundra.

The arable Gleyic Stagnosol plot is at a distance of 120 m to the north-west from the virgin Gleyic Stagnosol plot. The agricultural revegetation of the tundra took place in 1940s, in order to create artificial meadows [28]. For over 30 years, the soil was exposed to processing—the natural tundra vegetation was mulched, and following the introduction of organic and mineral fertilizers, crops were planted. Such lengthy agricultural use transformed the typical tundra soils into bluegrass and foxtail meadows. The agricultural exposure activated the sod-forming process, improved the soil water-air and agrochemical properties, and changed the temperature regime considerably.

All soils were sampled in triplicate. The morphological descriptions and physicochemical properties of the studied soils are presented in Table 2. More detailed descriptions were published earlier [25,29,30].

**Table 2.** The morphological descriptions and the physicochemical properties of the soils.

| Horizon | Depth, cm | Soil Horizon Description | pH $_{H_2O}$ | TOC, % | Sum of Particles < 0.01 mm, % |
|---|---|---|---|---|---|
| | | Taiga zone | | | |
| | | virgin Eutric Albic Histic Retisol | | | |
| He | 0–8 | undercomposed with fresh organic remnants | 4.2 | 34.3 ± 1.2 | n.d. |
| Eh | 12–20 | loamy, friable, penetrated by vertical cracks with brown humus impregnation of the walls | 4.8 | 0.77 ± 0.18 | 18.9 |
| E | 20–28 | loamy, dense, structureless | 5.0 | 0.35 ± 0.07 | 32.6 |
| E | 28–37 | loamy, dense, structureless | 5.1 | 0.23 ± 0.05 | 25.7 |
| | | virgin Eutric Albic Retisol | | | |
| Oe | 0–5 | friable organic material without histic features | 5.3 | 35.4 ± 1.2 | n.d. |
| Eh | 5–7 | loamy, friable, contains many roots and humus cutans | 4.1 | 1.8 ± 0.4 | 19.0 |
| E | 7–13 | loamy, friable | 5.1 | 0.46 ± 0.09 | 20.6 |
| E | 13–35 | loamy, slightly compacted, few roots | 5.1 | 0.35 ± 0.07 | 24.2 |
| | | arable Eutric Albic Retisol | | | |
| A$_P$ | 0–15 | light loamy, inclusions of peat, small concretions, loose-cloddy, many roots | 6.6 | 1.9 ± 0.4 | 25.4 |
| | | Tundra zone | | | |
| | | virgin Histic Cryosol | | | |
| Hi | 0–10 | histic undercomposed material | 3.7 | 27.9 ± 2.8 | n.d. |
| He | 10–20 | histic slightly composed material | 3.8 | 31.3 ± 1.1 | n.d. |
| Bfg | 28–41 | overmoisted loamy | 4.0 | 0.46 ± 0.09 | 35.6 |
| | | virgin Gleyic Stagnosol | | | |
| Oe | 0–5 | undercomposed litter | 5.6 | 18.4 ± 1.8 | n.d. |
| Bh | 5–10 | loamy, contain roots | 5.0 | 0.50 ± 0.10 | 25.0 |
| Bg | 10–30 | loamy, structureless | 5.3 | 0.29 ± 0.06 | 31.1 |
| | | arable Gleyic Stagnosol | | | |
| A$_P$ | 0–5 | slightly decomposed plant material on the soil surface (0–2 cm), loam in the lower part of the horizon (2–5 cm). The structure is lumpy-powdery, root interlacing | 5.3 | 27.3 ± 2.7 | n.d. |
| ABg | 5–20 | loamy, foliose structure, grass roots | 5.4 | 2.9 ± 0.6 | 33.8 |
| Bg | 20–35 | loamy, ferruginous concretions | 5.3 | 0.48 ± 0.10 | 30.7 |

Note: TOC – total organic carbon; n.d.—not determined.

## 3. Methods

### 3.1. Soil Analysis

Air-dried peat soil samples were homogenized and sieved through a 2 mm sieve. The total organic carbon (TOC) content was determined using an element analyzer EA-1110 (Carlo-Erba, Cornaredo, Italy) in the Chromatography Common Use Center (Institute of

Biology, Syktyvkar, Russia) and pH in water suspensions was determined using a pH-meter Hanna HI 8519 (Hanna Instruments, Vöhringen, Germany).

### 3.2. Extraction of HSs

HS samples were collected from organogenic and mineral soil horizons. HAs and FAs were obtained from air-dried soil samples by double extraction with a solution of 0.1 M NaOH (1:10 soil to solution ratio), according to the IHSS recommendations [31]. Further on, a saturated $Na_2SO_4$ solution (20% of the extract volume) was added to the alkaline extract to coagulate the colloidal particles and centrifuged for 1 h at 13,000 rpm. The HAs were precipitated in the purified extract by the gradual addition of a 10% $H_2SO_4$ solution, thus bringing the solution pH to 2. FAs was purified on activated carbon (AG-3 grade) and desalted by passing through a KU-2 cation exchanger in the H+ form [32]. To purify the HA and FA samples, they were exposed to dialysis, then transferred to porcelain dishes and dried in an oven at a temperature of 35 °C.

### 3.3. The EPR Measurements of the HSs

The EPR spectra were recorded on a JES FA 300 spectrometer (JEOL, Tokyo, Japan) in the X band at room temperature, and a microwave power of 1 mW, and a high frequency modulation amplitude of 0.06 mT. Due to manganese with a known concentration of free radicals serving as an external standard, it was introduced into the resonator of the EPR spectrometer. The content of the paramagnetic centers in the samples was determined through the comparison of the relative intensities of the signals of the standard and the sample; the calculation was performed in JES-FA swESR v. 3.0.0.1 (JEOL, Tokyo, Japan). The relative error of the EPR method determining the concentration of free radicals does not exceed 10% [33].

### 3.4. Statistical Analyses

Bivariate correlation analyses were conducted using the Pearson product-moment correlation coefficient (*r*), and its statistical significance was assessed via the Neyman–Pearson approach (normal distribution). Technically, the observed value of the coefficient (based on *n* pairs) was compared against the critical value ($r_{cr}$) for a two-tailed test and significance level ($\alpha$) of 0.05. The principal component analysis (PCA), using Statistica v. 12.1 (Dell, Round Rock, USA), was performed to determine the correlations between paramagnetic and molecular parameters of HAs and FAs. The number of the factors extracted from the variables was determined by the Kaiser rule. With this criterion, the first two principal components with an eigenvalue greater than two were retained [34]. All statistical estimations were performed with the predetermined significance level of $p \leq 0.05$.

## 4. Results and Discussion

### 4.1. The EPR Data of the HSs

Figure 2 shows an EPR spectrum typical for all studied HAs and FAs recorded within a relatively narrow range in the region of signals from the organic FRs. An intensely broad line with the g-factor of 2.0049 to 2.0063 was detected on the spectra of the samples under study, which indicates the presence of FRs in the HS structure (Table 3).

In terms of value, the obtained g-factor is close to the g-factor of a free electron (g = 2.0023), which supports the existence of a strongly delocalized molecular orbital in the HS structure [17,35]. However, it should anyway be noted that the orbital component of the electron's magnetic moment is not at zero, which leads to a growing g-factor.

The g-factor for the FA samples is averagely higher than the one for the HAs (Table 3), which indicates a greater shift in the unpaired electron density in the FA structure towards the oxygen atom, as a result of its spin-orbit interaction with various oxygen-containing functional groups, including semiquinone structures. According to some researchers, the g-factor values of 2.0040 and higher may serve to evidence the presence of ether or heterocyclic oxygen atoms of the furan structure type in the conjugation system of HAs [36].

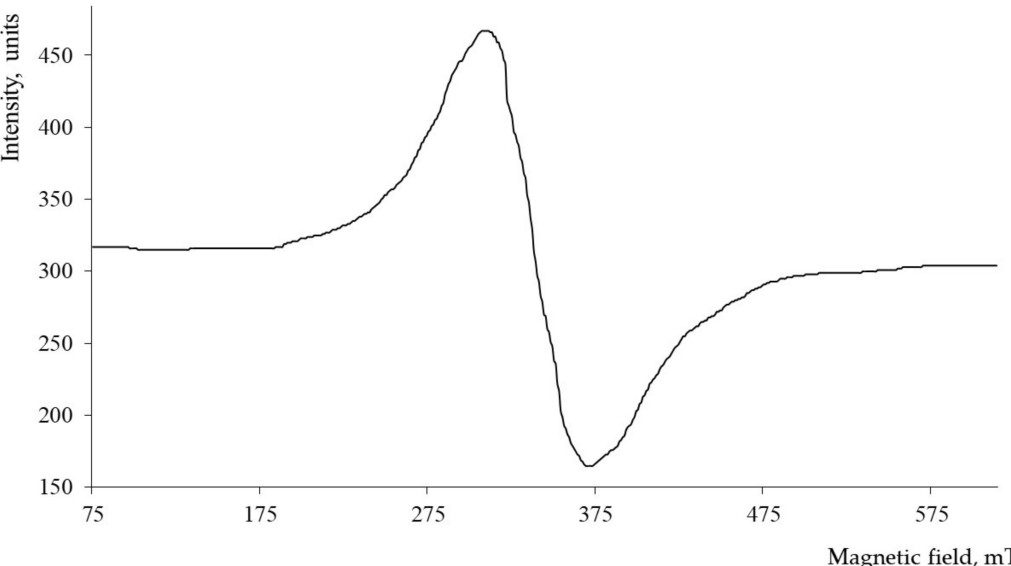

**Figure 2.** A typical EPR spectrum of the HSs (within the region of the g-factor of 2.003), based on a HA sample from the Eh horizon of Eutric Albic Histic Retisol.

**Table 3.** Paramagnetic properties of humic and fulvic acids.

| Horizon | Depth, cm | Humic Acids | | Fulvic Acids | |
|---|---|---|---|---|---|
| | | Free Radical Concentration, $\times 10^{15}$ Spin/g | g-Factor | Free Radical Concentration, $\times 10^{15}$ Spin/g | g-Factor |
| | | Taiga zone | | | |
| | | virgin Eutric Albic Histic Retisol | | | |
| He | 0–8 | 7.75 ± 0.47 | 2.0057 | 1.34 ± 0.08 | 2.0059 |
| Eh | 12–20 | 8.02 ± 0.48 | 2.0057 | 3.07 ± 0.18 | 2.0060 |
| E | 20–28 | 6.83 ± 0.41 | 2.0051 | 2.75 ± 0.17 | 2.0056 |
| E | 28–37 | 6.43 ± 0.39 | 2.0052 | 2.84 ± 0.17 | 2.0055 |
| | | virgin Eutric Albic Retisol | | | |
| Oe | 0–5 | 3.24 ± 0.19 | 2.0053 | 2.17 ± 0.13 | 2.0063 |
| Eh | 5–7 | 5.26 ± 0.32 | 2.0051 | 1.05 ± 0.06 | 2.0055 |
| E | 7–13 | 4.06 ± 0.24 | 2.0049 | 0.96 ± 0.06 | 2.0056 |
| E | 13–35 | 3.74 ± 0.22 | 2.0050 | 0.80 ± 0.05 | 2.0055 |
| | | arable Eutric Albic Retisol | | | |
| $A_p$ | 0–15 | 4.05 ± 0.22 | 2.0055 | 0.50 ± 0.03 | 2.0062 |
| | | Tundra zone | | | |
| | | virgin Histic Cryosol | | | |
| Hi | 0–10 | 2.60 ± 0.13 | 2.0052 | 0.48 ± 0.03 | 2.0065 |
| He | 10–20 | 1.61 ± 0.10 | 2.0059 | 0.26 ± 0.02 | 2.0055 |
| Bfg | 28–41 | 3.90 ± 0.20 | 2.0053 | 0.98 ± 0.06 | 2.0059 |
| | | virgin Gleyic Stagnosol | | | |
| Oe | 0–5 | 3.42 ± 0.17 | 2.0052 | 2.21 ± 0.11 | 2.0050 |
| Bh | 5–10 | 2.73 ± 0.14 | 2.0049 | 0.56 ± 0.03 | 2.0054 |
| Bg | 10–30 | 5.92 ± 0.30 | 2.0050 | 1.50 ± 0.08 | 2.0053 |
| | | arable Gleyic Stagnosol | | | |
| $A_p$ | 0–5 | 0.49 ± 0.03 | 2.0059 | 0.18 ± 0.02 | 2.0057 |
| ABg | 5–20 | 2.40 ± 0.12 | 2.0057 | 1.11 ± 0.06 | 2.0057 |
| Bg | 20–35 | 6.20 ± 0.31 | 2.0055 | 1.78 ± 0.10 | 2.0057 |

Note: the absolute error for the determination of the g-factor is 0.0002.

The calculations of the integral intensity of the absorption line enabled us to estimate the concentrations of the unpaired electrons in the HS samples. It has been established that the concentration of the paramagnetic centers in the HA samples is several times

higher than in the FAs, which is associated with a higher content of aromatic and other polyconjugated structures in the HA molecules, where unpaired electrons can delocalize, and is declarative of a higher capacity of HAs to undergo free radical polymerization and complex formation.

As can be seen in Table 3, an increased moisture content in Retisols leads to a higher content of FRs in the molecular structure of HAs from all of the horizons. It is explained by the biohydrothermal conditions of the humus formation in Eutric Albic Histic Retisol, under which all biochemical processes are inhibited [30,33], and the plant litter humification degree is low, which contributes to the formation of HSs with a high content of FRs in their structure.

The highest concentration of FRs in HAs was noted in the organo-mineral horizons (Eh) of the virgin Retisols, which may be due to an increased proportion of quinone fragments in the structure of HAs and a greater stabilization of unpaired electrons. This fact is implicitly supported by a significantly high correlation coefficient ($r = 0.911$; at $p = 0.95$, $n = 5$, and $r_{cr} = 0.878$) between the signal intensity of the $^{13}$C-NMR spectra in the range of 183–190 ppm [25] and the FR concentration in the HAs under study.

The FA samples show a different picture of the changing FR concentration across the profiles of the soils under study (Table 3). Thus, the FAs from virgin Eutric Albic Retisol tend to at least halve the content of FRs in the mineral horizons. That means that for automorphic Eutric Albic Retisol, the FA samples from the mineral horizons are more thermodynamically stable, than the ones from the organogenic horizon. The decrease in the concentration of FRs, down the profile, is possibly due to their interaction with transition metal ions in the reduced form ($Fe^{2+}$, $Mn^{2+}$, etc.) in an acidic medium [37].

The Eutric Albic Histic Retisol, on the contrary, reveals an increased FR content in the FA structure, when moving from the organogenic He horizon to the transitional Eh horizon, followed by its slight decrease down the profile. It may be associated with the flushing regime of Eutric Albic Histic Retisol and the downward migration of water-soluble FAs across the profile.

The increased moisture level in the tundra soils leads to a lower content of paramagnetic centers in HAs and FAs (Table 3). The data on the molecular fragment concentration ($^{13}$C-NMR spectroscopy) [38] established a high correlation between the aromaticity degree in the HA of tundra soils and the FR content ($r = 0.746$), as well as a close to significant correlation between the mass fraction of the quinone functional groups and the g-factor $r = 0.588$ ($n = 8$, $p = 0.95$, $r_{cr} = 0.707$). With regard to the FA samples, the respective correlation ratios $r = 0.712$ and $0.582$ are non-significant ($n = 5$, $p = 0.95$, $r_{cr} = 0.878$). Therefore, the transformation of molecules accompanied by an increase in the proportion of thermodynamically stable aromatic structures should be considered the key reason for the growth of the paramagnetic activity of the tundra HSs. The data on the content of the paramagnetic centers of the HA and FA molecules are in greater agreement with the molar ratio $x$(H):$x$(C). For the HAs, the correlation ratio value is $r = 0.834$ ($n = 8$, $p = 0.95$, $r_{cr} = 0.707$), and for FAs $r = 0.901$ ($n = 6$, $p = 0.95$, $r_{cr} = 0.814$). It was with the increased condensation degree that Kononova [39] associated the high FR concentration in the HAs of Chernozems, compared to the HAs of the Retisols. Thus, the share of aromatic structures in the HS molecules turns out to be the main factor connected with the paramagnetic activity of HSs.

The difference in the character of changes in the paramagnetic activity of the HAs in virgin taiga and tundra soils, as a result of a higher moisture content, may be explained by a different nature of plant residues incoming and participating in the humus formation process. Namely, in taiga soils, with a higher degree of hydromorphism in spruce phytocenoses in blueberry-moss to long-moss-sphagnum forests, the share of tree litter increases from 46 to 76% and the share of moss-lichen litter decreases from 23 to 12% (in terms of the carbon from the total phytomass) [40], which results in a higher concentration of lignin structures involved in the formation of HSs and contributing to the stabilization of FRs. The ground cover of tundra soils is dominated by moss and lichen vegetation, and the increasing moisture enhances their proportion. In addition to the fact that vascular plants

contain lignin structures almost an order of magnitude higher than mosses and lichens do [27,41,42], we observe a decrease in the paramagnetic activity of HAs when shifting from Gleyic Stagnosol to Histic Cryosol.

The paramagnetic properties of HSs are also determined by redox processes occurring in soils. The tundra zone soils display predominantly reducing conditions, due to their high moisture content, which is caused by anaerobic processes leading to gleying [43]. Under such conditions, the humification of plant litter decreases, contributing to the predominance of prohumic substances with a low content of FRs in the humus of the upper horizons, compared with taiga soils. In such regimes, paramagnetic centers of the semiquinoid type are formed through the reduction of quinones. Semiquinone is most stable in its radical ion form at higher pH values [17]. This may be one of the reasons for the decreased FR content in the Bh horizon, which has a lower pH value in virgin Gleyic Stagnosol. The reason for a greater stability of quinoid-type structures in the HA composition, which determines their higher content, is an increase in the pH values in the Bg mineral horizons of tundra soils. The reducing conditions in tundra soils lead to a decrease in the FR, compared to the taiga zone soils.

Table 3 shows that the content of the paramagnetic centers in HAs from the arable horizon (Ap) of Eutric Albic Retisol, is almost in congruence with the average concentration of FRs in the O and Eh horizons of virgin soils. In contrast to HAs, FAs from the arable Eutric Albic Retisol are less thermodynamically stable and more subject to microbial degradation. This, in its turn, explains a more intense transformation of the FA molecules in the course of agricultural use, resulting in the sharp decline of the FR concentration to trace amounts. The literature shows no agreement with regard to the changing FR concentration in HSs caused by soil reclamation.

The exploration of tundra soils creates more favorable microbiological and hydrothermal conditions, which contributes to a more rapid decomposition of organic matter. This is exhibited in a significant decrease in the free radical concentration in HAs in organogenic horizons.

According to some researchers, the concentration of paramagnetic centers in HAs is the reciprocal of their biothermodynamic stability and the depth of humification. When soils are plowed up, the paramagnetic activity of humic substances decreases [17,19]. Other authors believe that the presence of the remains of some agricultural crops in the humus composition can lead to a higher number of FRs [44].

The comparison of the obtained results with the literature data showed that the HA samples under study display a significantly lower paramagnetic activity, compared to the HAs of soils in the southern regions: the East European Plain—chernozems and sulfur forest soils [45], Pannonian Plain—Dystric Planosol, Eutric Cambisol, Orthic Luvisol and Stagno-Gleyic Cambisol [46], Silesian Lowlands—Cambic and Gleyic Podzols [18] and tropical soils of South America—Inceptisol, Alfisol, Entisol, Vertisol, Spodosol [47], Oxisol [35,48] and Gley [49], which is likely to be associated with the higher aromaticity of HAs in these soils.

In general, studies of paramagnetic activity of HSs of the cold soils are few, if any. FR concentrations obtained for HSs of the soils under study approximate HAs extracted from the soils of Bolshoy Lyakhovsky Island and Wrangel Island in the Arctic [16,50].

### 4.2. Statistical Analyses

The specific compositional parameters of HAs and FAs from taiga and tundra soils provided by the individual chemical and spectroscopic assays, were further subjected to statistical analysis using the PCA method. The results of the PCA explained 78.40% of the total variability of the HA properties isolated from Retisols (taiga zone). The dimension of the 14 input variables was reduced by PCA to two principal components with eigenvalues higher than two: the first axis (PC1) explained 53.68%, and the second (PC2) 24.72% of the total variability, while the third axis (PC3) explained 12.75% (Table 4).

**Table 4.** Analysis of the principal components for the HS parameters.

| Principal Components | Eigenvalues | % of Total Variance | Cumulative Eigenvalues | Cumulative % of Variance |
|---|---|---|---|---|
| *Taiga soils* | | | | |
| *Humic acids* | | | | |
| PC1 | 7.52 | 53.68 | 7.52 | 53.68 |
| PC2 | 3.46 | 24.72 | 10.98 | 78.40 |
| PC3 | 1.78 | 12.75 | 12.76 | 91.14 |
| *Fulvic acids* | | | | |
| PC1 | 7.80 | 55.68 | 7.80 | 55.68 |
| PC2 | 3.93 | 28.08 | 11.73 | 83.76 |
| PC3 | 2.27 | 16.24 | 14.00 | 100.00 |
| *Tundra soils* | | | | |
| *Humic acids* | | | | |
| PC1 | 5.98 | 42.73 | 5.98 | 42.73 |
| PC2 | 3.52 | 25.12 | 9.50 | 67.86 |
| PC3 | 2.90 | 20.75 | 12.40 | 88.60 |
| *Fulvic acids* | | | | |
| PC1 | 5.92 | 53.84 | 5.92 | 53.84 |
| PC2 | 2.47 | 22.45 | 8.39 | 76.28 |
| PC3 | 1.81 | 16.46 | 10.20 | 92.74 |

The PC1 was positively associated with the FR concentration (c(FRs)) and the share of quinone fragments, g-factor, atomic ratios H/C and C/N of the HAs and negatively coordinated with the share of O,N-arom, C,H-arom, O,N-alkyl fragments and the O-CH$_3$ groups of HAs on this axis. The PC2 was negatively related to c(FRs), g-factor, the share of quinone and the carbonyl fragments and atomic ratios O/C and C/N of the HAs (Figure 3A).

The projection of the taiga FA parameters (Figure 3B), using PCA, demonstrates that the PC1 is positively associated with the g-factor and the share of O,N-arom, C,H-arom, O,N-alkyl fragments and the O-CH$_3$ groups of FAs and is negatively related to the content of the oxygen-containing functions of FAs: O/C atomic ratio, carboxyl, carbonyl and quinone groups.

The PCA analysis of the HAs from tundra soils shows a statistical relationship between the g-factor and the proportion of the O,N-arom fragments, O-CH$_3$ groups and C/N atomic ratio (PC 1); c(FRs) is related to the proportion of C,H-arom (PC2) (Figure 3C), and the relationship between the g-factor and the proportion of quinone fragments (PC3). Projection of the FA (tundra soil) parameters using PCA, demonstrates that the PC2 is negatively associated with the g-factor and the quinone fragments and the O-CH$_3$ groups of FAs (Figure 3D).

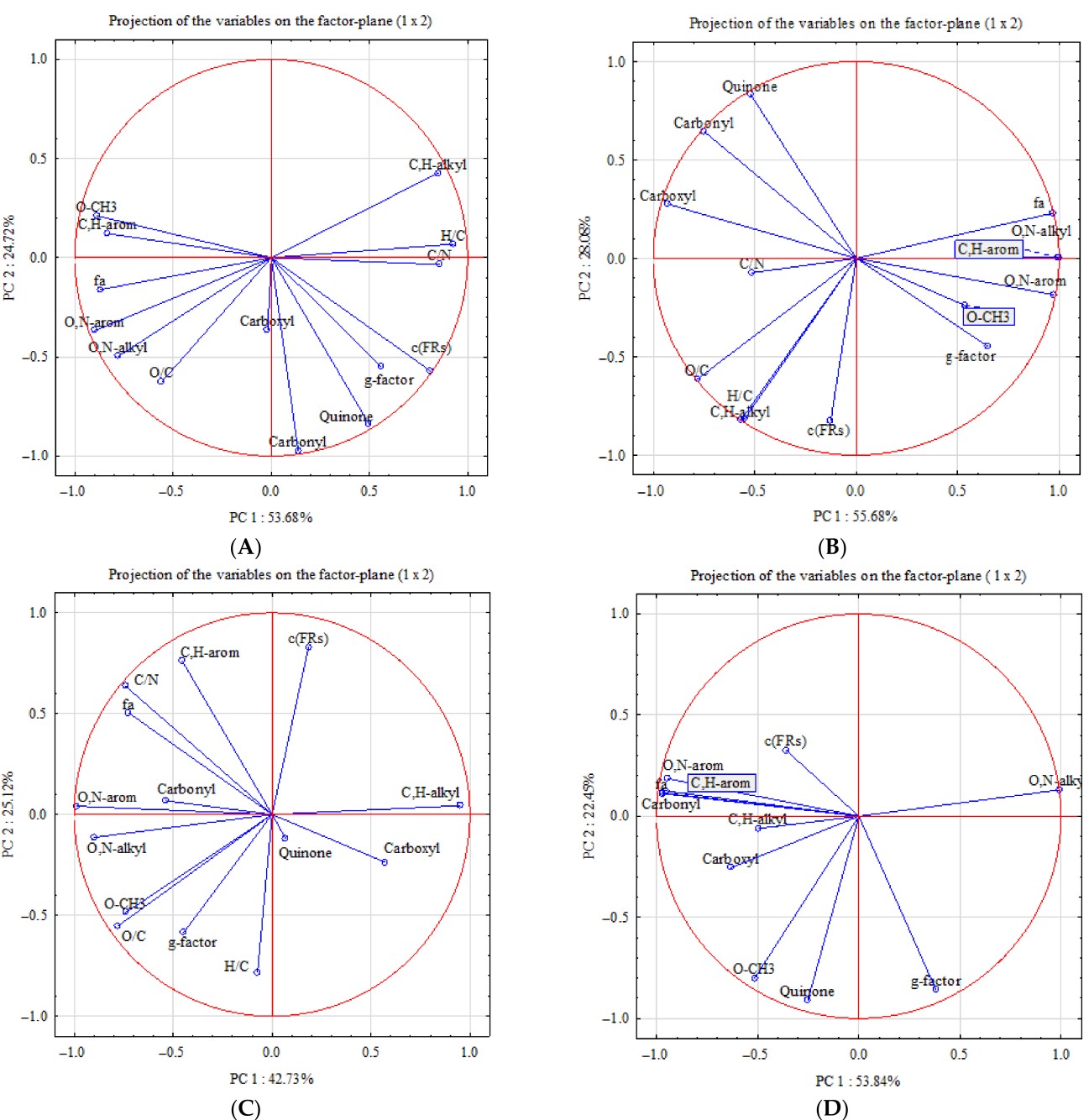

**Figure 3.** Projection of the HSs parameters using PCA: taiga HAs (**A**) and FAs (**B**); tundra HAs (**C**) and FAs (**D**).

## 5. Conclusions

The evaluation of the paramagnetic activity of the HS samples obtained from the taiga and tundra soils showed that the FR concentration in the HA samples is several times higher than the one in the FAs. It is explained by a higher content of aromatic and other polyconjugated structures in the HA molecules, on which unpaired electrons can delocalize. The PCA analysis of the HSs shows a statistical relationship between c(FRs) and the proportion of aromatic fragments, which indicates the active participation of these structural units in the radical polymerization and complexation reactions.

The value of the g-factor for the FA samples is averagely higher than the one for HAs. This indicates a greater shift of the electron density of the unpaired electron in the FA structure towards the oxygen atom, due to its spin-orbit interaction with oxygen-containing

functional groups, which demonstrate a higher content in FAs, compared to HAs. A statistically significant relationship was established between the value of the g-factor and the content of the O-CH$_3$ groups in FAs, which can be associated with lignin fragments.

An increase in hydromorphism in the Retisols leads to a higher concentration of FRs in the molecular structure of HAs. It may be explained by the different nature of the input and participation of plant residues enriched in the lignin components in the process of the humus formation.

A higher moisture content in tundra soils leads to a lower content of free radicals in the HS structure, which is due to the high proportion of moss species in plant residues.

In contrast to HAs, no clear regularity is observed for the FAs in their paramagnetic activity when analyzing the profiles of the studied soils. It may be associated with a good solubility of FAs and their subsequent mixing in the course of a downward migration across the profile.

The agricultural use of taiga and tundra soils reduces the paramagnetic activity of HSs by over 20%, compared to their virgin counterparts, which is due to a higher rate of transformation of plant precursors and prohumus substances in the arable horizon.

Information about the molecular structure and paramagnetic properties of humic substances is necessary to assess the supply of plants with nutrients in the soil, including phosphorus, iron, etc., and can be used to directly influence the growth of higher plants.

**Author Contributions:** Conceptualization, E.L.; methodology, E.L.; software, R.V. and E.A.; validation, E.L., R.V., and E.A.; formal analysis, E.L., R.V., and E.A.; investigation, E.L.; resources, E.L., R.V. and E.A.; data curation, E.L. and R.V.; writing—original draft preparation, E.L. and R.V.; writing—review and editing, E.L. and E.A.; visualization, E.L. and R.V.; supervision, E.L.; project administration, E.L.; funding acquisition, E.L. and R.V. All authors have read and agreed to the published version of the manuscript.

**Funding:** The reported study by Lodygin E. and Vasilevich R. was funded by the Federal budget of Russia, within the framework of the research topic of the Institute of Biology (No. 122040600023-8). The study by E. Abakumov was carried out within the framework of the Russian foundation for basic research project grant No 19-05-50107.

**Institutional Review Board Statement:** Not applicable.

**Informed Consent Statement:** Not applicable.

**Data Availability Statement:** Not applicable.

**Conflicts of Interest:** The authors declare no conflict of interest. The funders had no role in the design of the study; in the collection, analyses, or interpretation of the data; in the writing of the manuscript; or in the decision to publish the results.

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
