# Peer review of "Characterization of Humic Substances from Taiga and Tundra Soils by EPR Spectroscopy"

_agronomy, doi:10.3390/agronomy12112806_

Round 1

Reviewer 1 Report

Electron paramagnetic resonance spectroscopy, which can be used to detect qualitatively and quantitatively the unpaired electrons contained in atoms or molecules of matter and to explore the structural properties of their surroundings. The paper is well conceived and has an adequate amount of data, but there are still some problems.

1.        The background and shortcomings of the study in the abstract of the paper are not clear, nor is the significance of the study explicit.

2.        The importance of HS and FAs in soils is highlighted in the Introduction, but less integrated with production practice.

3.        Tabel 2 has different sampling depths for different Horizon, which will have an effect on Humic Acids and Fulvic Acids; the data are not compared significantly; soil moisture has a great influence on EPR values, is the sample dry or wet?

4.        Tabel1 only gave the Soil Name and Vegetation Type, but different soil types have different adsorption capacity for Humic Acids and Fulvic Acids, so the soil type should be clearly explained.

5.        the analysis was conducted only from EPR data, not verified with directly related data, especially the lack of FA, HA, and organic carbon data, and no direct connection between different types of soil humification and FRs was established, which appeared to be a lack of research significance.

6.        Many expressions in the conclusion are speculations, which also indicate that some extraneous factors were not considered.

Author Response

Dear reviewer, we are cordially thanks you for detailed work with paper and providing suggestions and recommendations for improving the manuscript text.
All changes in the text we marked yellow.
Detailed comment for all suggestions provided below.

1. Annotation corrected.
2. Introduction expanded and corrected.
3. Different types of soils have different thicknesses of horizons. We selected samples by horizons, which allows us to correctly compare the properties of HA and FA preparations. The HS samples were isolated from air-dry soil samples. This is stated in chapter 3.1. Extraction of HSs.
4. The name of the soil and reflects the type of soil. For a better understanding of soil properties, Table 2 has been added.
5. Previously, these preparations of HAs and FAs were studied by us by 13C-NMR spectroscopy and the results were published: https://doi.org/10.3390/agronomy12010144, https://doi.org/10.24425/ppr.2020.133009 https://doi. org/10.1134/S1064229314010074. The same articles also provide elemental analysis of HA and FA. Therefore, we did not duplicate them. The PCI analysis available in our article is built using these data.
6. We have tried to ensure that the conclusion contains confirmed data.

With Kind Regards,
Authors

Reviewer 2 Report

General Comments

The topic of current study is suitable for AGRONOMY journal. The paper is dealing with, and may help to better understand, differences in soil organic matter properties in virgin and arable land soil in taiga and tundra. Study have been conducted using EPR spectroscopy of soil OM humic fraction extracted from soils in the natural habitats or agricultural lands. Six sampling sites in two twrritories were selected.

Soil organic matter and OM persistence has been the topic of intense study for the last decades. There are ongoing debate on soil organic C dynamics.  On the one hand 2011 Michael Schmidt et al. published their paper "Persistence of soil organic matter as an ecosystem property" (Nature, 2011. volume 478, pages 49–56), on the other hand,  Gerke, J., 2018. Concepts and misconceptions of humic substances as the stable part of soil organic matter: a review (Agronomy, 8, 76).  It would be useful to discuss how these results come within the debate on soil orgnanic matter persistence, does these results have impact on knowledge on soil OM chemical persistence.

Specific comments:

1). Abstract

Could the authors highlight the results and their aim in soil science?

2) Introduction section:

Well written.

3) Field sampling

L92 – L 93. There are mentioned, that soil physical properties are described in previous papers. Could authors provide information on soil basic physical properties and add to Table 1. It would help determine that soils are comparable within one study region.

3). Results and Discussion

L167 – L170. Could the authors provide the explanation or discuss the possible mechanisms behind the Fr changes across soil profile.

L236 – L246. The described humification conditions (moisture, plant residues – lichens) in tundra soils are comparable with conditions in raised bogs. Is there any papers on HS paragmagnetic activity in peat bogs?

Figure 4. My suggestion is to add data points, that will help reader better visualize soil samples with HS parameters.

4) Conclusions

L289 – L295. Both paragraphs states that increase in hydro morphism leads to higher FR in one case or to lower FR in another case. Both processes are explained with composition in plant residues. It is confusing. Please improve this part of the conclusions.

Author Response

Dear reviewer, we are cordially thanks you for detailed work with paper and providing suggestions and recommendations for improving the manuscript text.
All changes in the text we marked yellow.
Detailed comment for all suggestions provided below.

1) Abstract and Introduction added and corrected

2) Added table 2 with physical and chemical properties of soils.

3). A possible mechanism for decreasing the concentration of FRs down the profile has been added to the text.

4) Yes, the conditions of humification in tundra soils are comparable to those in peatlands. However, we have not seen such publications. We have our own data on the paramagnetic activity of HSs from peat soils, but they have not yet been published.

5) Statistica 12 features do not allow you to add data points to the same figure. We believe that adding four more separate drawings will increase the size of the manuscript..

6) Conclusions was corrected.

With Kind Regards,
Authors

Reviewer 3 Report

See attached file

Author Response

Dear reviewer, we are cordially thanks you for detailed work with paper and providing suggestions and recommendations for improving the manuscript text.
All changes in the text we marked yellow.
Detailed comment for all suggestions provided below.

1) The purpose has been adjusted

2) Full reference to the World Reference Base for Soil Resources 2014 is provided in the Reference.

3). The boundary between the taiga and tundra zones is not a straight line and its mapping is quite difficult. The studied tundra soils are located north of the Arctic Circle, as can be seen from the figure.

4) For the extraction of HSs, we used the author's methods. Links to them are provided in the text.

5) Table 2 (The morphological descriptions and physicochemical properties of the soils) has been added to the text. The morphological descriptions and physicochemical properties of the soils.

6) Yes, this bug has been fixed.

7) Unfortunately, we don't have good photos of all sections and landscapes.

With Kind Regards,
Authors

Round 2

Reviewer 1 Report

Thank you .

No other issues.

Reviewer 3 Report

The authors have followed most of the recommendations given. Therefore, as a consequence of the changes and corrections incorporated in this new version, the scientific quality of the manuscript has considerably improved. Then, it can be accepted.